# Interseismic strain build-up on the submarine North Anatolian Fault offshore Istanbul

Dietrich Lange [1,11], Heidrun Kopp [1,2,11], Jean-Yves Royer [3], Pierre Henry[4], Ziyadin Çakir [5], Florian Petersen [1], Pierre Sakic [6,7], Valerie Ballu[6], Jörg Bialas [1], Mehmet Sinan Özeren[8], Semih Ergintav[9] & Louis Géli [10]

Using offshore geodetic observations, we show that a segment of the North Anatolian Fault in the central Sea of Marmara is locked and therefore accumulating strain. The strain accumulation along this fault segment was previously extrapolated from onshore observations or inferred from the absence of seismicity, but both methods could not distinguish between fully locked or fully creeping fault behavior. A network of acoustic transponders measured crustal deformation with mm-precision on the seafloor for 2.5 years and did not detect any significant fault displacement. Absence of deformation together with sparse seismicity monitored by ocean bottom seismometers indicates complete fault locking to at least 3 km depth and presumably into the crystalline basement. The slip-deficit of at least 4 m since the last known rupture in 1766 is equivalent to an earthquake of magnitude 7.1 to 7.4 in the Sea of Marmara offshore metropolitan Istanbul.

[1] GEOMAR Helmholtz Centre for Ocean Research Kiel, Kiel, Germany. [2] University of Kiel, Kiel, Germany. [3] Laboratoire Géosciences Océan, Université de Brest and CNRS, Plouzané, France. [4] Aix Marseille Univ, CNRS, IRD, INRA, Coll France, CEREGE, Aix-en-Provence, France. [5] Faculty of Mines, Department of Geology, Istanbul Technical University, Istanbul, Turkey. [6] Laboratoire LIENSs, Université de la Rochelle and CNRS, La Rochelle, France. [7] GFZ Helmholtz-Zentrum Potsdam, Potsdam, Germany. [8] Eurasia Institute of Earth Sciences, Istanbul Technical University, Istanbul, Turkey. [9] Kandilli Observatory and Earthquake Research Institute, Department of Geodesy, Bogazici University, Istanbul, Turkey. [10] Institut Français de Recherche pour l'Exploitation de la Mer (IFREMER), Département Ressources Physiques et Ecosystèmes de Fond de Mer, Unité des Géosciences Marines, Plouzané, France. [11] These authors contributed equally: Dietrich Lange, Heidrun Kopp. Correspondence and requests for materials should be addressed to D.L. (email: dlange@geomar.de)

It is well known that Istanbul city and populations along the coasts of the Sea of Marmara were previously severely affected by earthquakes related to the submerged North Anatolian Fault (NAF) in the Sea of Marmara[1]. Some of the earthquakes were associated with seismically driven sea-waves and six destructive run-ups are known from historical reports for the last 20 centuries[2]. For example, the 1766 earthquake, suggested to have nucleated beneath the western Sea of Marmara[3], resulted in very strong shaking in Istanbul (Mercalli Intensity VII, very strong shaking) and seismically driven sea-waves submerged the quays in Istanbul[2].

The Sea of Marmara, crossed by the NAF, is one of the regions on the globe where the fragmentary knowledge on the degree of fault locking poses a significant impediment for assessing the seismic hazard in one of Europe's most populated regions, the Istanbul metropolitan area. Since 1939, destructive seismic events on the onshore portion of the NAF have propagated westwards towards Istanbul (Fig. 1c)[4]. The most recent events were the Mw 7.2 Düzce and Mw 7.4 Izmit earthquakes in 1999 (Fig. 1a) that caused 854 and ~18,000 casualties, respectively. Towards the Dardanelles in the west, the Mw 7.4 Ganos earthquake ruptured

the NAF in 1912[5]. In the Sea of Marmara, the NAF forms a well-known seismic gap along a 150 km-long segment[6], inferred to have last ruptured in 1766[1], whereas all the onshore segments of the NAF from the province Erzincan in Eastern Anatolia to the Sea of Marmara ruptured in the last 100 years[7]. The degree of aseismic deformation and hence the locking state of the marine fault segments of the NAF cannot be well resolved using onshore GPS stations alone[8]. Owing to the lack of offshore observations, the uncertainty on fault slip rates on the order of 10 mm a$^{-1}$ in the central part of the Sea of Marmara prevails.

While onshore deformation of faults is monitored using geodetic techniques such as GPS and InSAR, movement of offshore faults remains mostly unknown due to the opacity of water to electromagnetic waves. As a result, we rely on extrapolated observations of GPS land measurements to the marine domain[9,10]. However, extrapolating onshore observations requires assumptions about crustal properties and fault geometry to determine the locking state of a fault or fault segment[8]. Inferring the slip rate of faults from seismicity[11-13] includes assumptions about the frictional behaviour of the fault because deformation is known to be partitioned into seismic moment

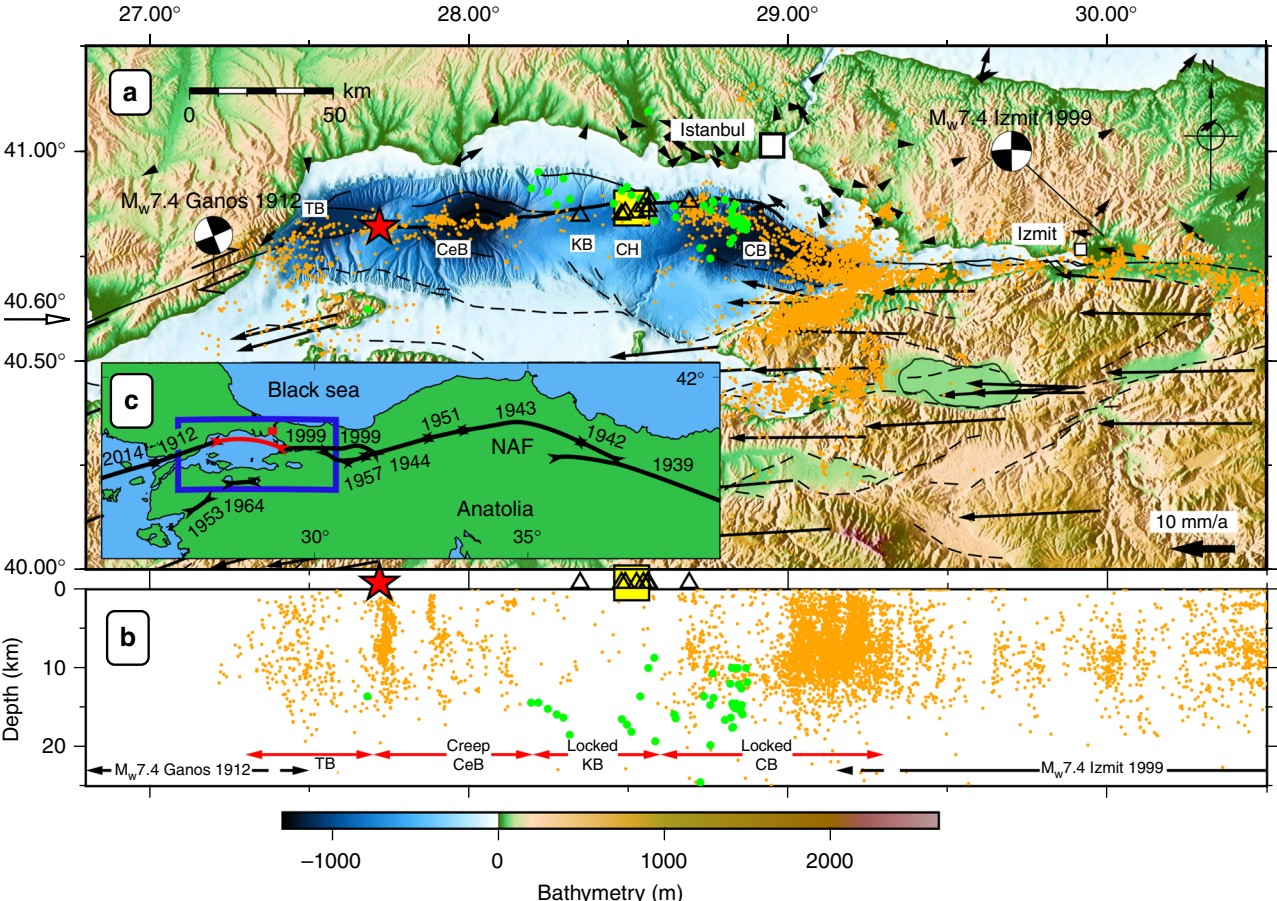

**Fig. 1** Overview and tectonic setting of the Sea of Marmara. **a** Tectonic setting of the NAF (solid line) in the Marmara region with local seismicity (orange dots) between 1999–2009[4] and 2010–2012[11]. The geodetic acoustic network is located in the yellow box (top center) and the local ocean bottom seismometer (OBS) stations (29/10/2014–25/04/2015 and 26/04/2015-13/04/2016) are indicated with triangles. Microseismicity based on the OBS (this study) in the area of the geodetic network is shown with green circles. The red star indicates the location of a recent Turkish-Japanese direct path-ranging network in the western Sea of Marmara[22]. Fault traces of the NAF[29] and GPS displacements relative to stable Eurasia[9] are shown with black arrows and lines. Bathymetry from[30] and topography from[31]. Tekirdağ basin (TB), Central Basin (CeB), Kumburgaz Basin (KB), Central High (CH) and Çınarcık Basin (CB)[11]. **b** Profile view of seismicity north of 40.6°N, same symbols as in panel a. Sedimentary basins are indicated with red arrows and the extent of the Ganos 1912 and Izmit 1999 earthquakes[12] are indicated with black arrows. Creeping[11,12,22] and locked[6] segments of the NAF are labelled. See text for discussion about locked fault beneath the KB and the CB. **c** Large-scale tectonic setting of north-western Turkey with rupture zones of major earthquakes[6,25] along the North Anatolian Fault (NAF)

release (e.g. through micro-seismicity) and aseismic creep[14]. Other studies use the relationship between the frequency and magnitude of earthquakes, known as the Gutenberg–Richter law, to infer the state of differential stresses on faults[15]. Our blindness to offshore deformation leads to pivotal debates in science. For instance, the kinematic state of a fault can vary between the two end members from fully locked to continuously creeping, resulting in a slip accumulation between zero and full displacement along the fault. The fault kinematics is thus determinative of hazard estimates and we are far from the level of knowledge we have on deformation onshore[16]. Seafloor uplift and subsidence can be resolved by pressure measurements[17] and absolute horizontal displacement by combining GPS and acoustic measurements[18]. Direct and continuous acoustic path-ranging between two sites on the seafloor becomes increasingly used in oil exploration[19] and research[20–23]. Recent works used submarine fibre optic telecommunication cables to detect subtle strain changes induced by distant earthquakes[24]. Here, we use acoustic ranging techniques to measure horizontal crustal strain on the seafloor with mm-precision over periods of years and dozens of baselines to resolve tectonic deformation. The geodetic monitoring together with the OBS observation indicate that the NAF segment in the Kumburgaz Basin is fully locked.

## Results

**Direct path ranging experiment.** Our inability to resolve subhorizontal tectonic displacement on the seafloor is addressed by a path-ranging method based on acoustic travel-time measurements[16,20–22,25]. Distance changes between seafloor instruments are estimated from two-way travel times and sound speed of water. Sound speed along the entire ray-path is approximated by the geometric mean of the sound velocities measured at both endpoints. In October 2014, an offshore geodetic network of intercommunicating transponders was installed in 800 m water depth where the NAF trace is identified in high-resolution multibeam seafloor bathymetry maps[26,27] acquired by autonomous underwater vehicles (Fig. 2) and in 3.5 kHz seismic profiles[28]. The deployment site was selected based on the existence of a linear scarp of the NAF clearly visible in the

bathymetry as a proxy for maximised strain release along the NAF[29,30]. Ten acoustic transponders, four from the Ocean Geosciences Laboratory in Brest, France (station names starting with F) and 6 from GEOMAR's GeoSEA array in Kiel, Germany (station names starting with G), were installed[25] in October 2014 and remained fully operational until May 5, 2017. Each set of stations, manufactured by Sonardyne Ltd (UK), used a different center-frequency (F: 22.5 kHz, G: 17 kHz), so the F- and G-stations communicated only with F- and G-stations, respectively. Pairs of stations, <100 m apart, shared common baselines and all stations monitored the temperature, pressure, and tilt (Fig. 3a, b, Supplementary Figs. 1–3) at the transponder site along with the two-way travel time between them (Fig. 3c).

**Baseline estimation.** We use 650.000 two-way travel time measurements based on two soundings, respectively. Each station either interrogates the other stations of its kind or acts as a replying instrument, thus the experiment forms an autonomous intercommunicating network on the seafloor, rather than observations at individual positions. The acoustic distance is then calculated from the average sound-speed in water multiplied by the one-way travel time.

We estimate the sound-speed in water from temperature and pressure measurements[32], assuming a constant salinity, similar to other path-ranging experiments[20,33,34]. Figure 3 shows the time series of the measured parameters together with the calculated sound speed and baselines for the southwest-northeast fault crossing baseline G2-G5.

For the final strain estimates (Fig. 4), we subtracted a mean strain of $4.5 \times 10^{-6}$ estimated from all baselines not crossing the fault (corresponding to a baseline decrease of 4.5 mm for a 1000 m long baseline during 2.5 years). Subtracting a constant strain value, instead of a constant increase or decrease in the baseline lengths, accounts for the fact that linear trends in sensor drift or water parameters are linearly related to strain. In addition, considering strain makes the observation independent from the baseline lengths. The baselines show a consistent behaviour (Fig. 2, Supplementary Figs. 4, 5 and Supplementary Table 1), with a long-term resolution better than 8 mm for all baselines

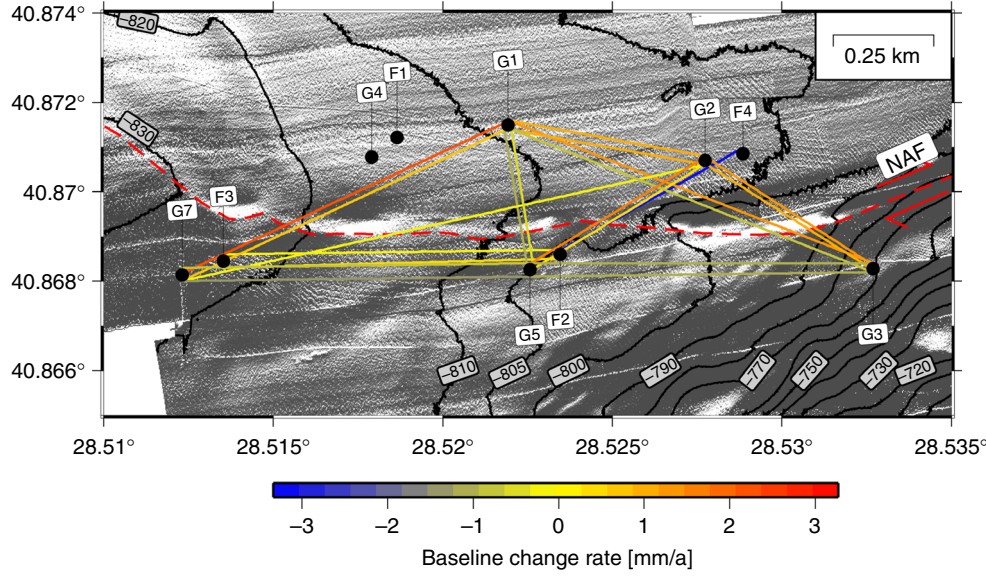

**Fig. 2** Map of the geodetic network on the seafloor. Baseline shortening or lengthening rates, based on the total deformation measured at the end of the deployment, are colour coded. A constant strain of $4.5 \times 10^{-6}$ was subtracted to each baseline in order to correct for the inferred constant salinity decrease of 0.002 PSU yr$^{-1}$. Transponder locations are shown as black circles. F- and G- stations only communicate within their respective network. High-resolution shaded bathymetry map[26,27] is shown with a light source from the South

(Fig. 3), and strain is smaller than $1 \times 10^{-5}$ for all baselines (e.g. less than 1 cm deformation on a 1 km-long measurement

distance). For instantaneous baseline changes, the resolution capability is ~2 mm, in particular, if an event would influence different baselines. We interpret the nominal lengthening of all baselines as a result of subtle changes in the physical properties of water, such as a salinity decrease of 0.005 practical salinity units (PSU) during the 2.5 years of deployment (0.002 PSU a$^{-1}$). Alternatively, one could consider a total drift in all pressure sensors of −0.242 kPa or a total drift in all temperature sensors of −0.00126 °C. In particular, a pressure drift of −0.242 kPa during the deployment cannot be distinguished from a local uplift of 2.4 cm. Because the Sea of Marmara is an extensional step-over or pull-apart structure system including substantial subsidence[35] with the basement imaged at a depth of ~4.5 km below the Central High and therefore close to the geodetic network, uplift is unlikely[36]. The data suggest an absence of vertical movement and the remaining small baseline changes originate from measurement uncertainties of the pressure and temperature sensors. The subtle changes of water parameters such as the temperature increase of 0.002 °C on March 2016 (Fig. 3a) might be a sensor artefact since they are not compensated by pressure or travel time resulting in an apparent ~3 mm baseline length change (Fig. 3e). However, since baseline estimates are based on the equation distance = time × velocity, travel times and water sound-speeds must be jointly accurately known. This problem is similar to the hypocenter-depth-velocity dependency in earthquake location techniques. This is the reason why the network was designed to measure a high number of baselines across the fault to allow isolating effects of sensor drift from baseline changes.

**Estimation of slip on the fault from baseline data.** We compared the observed baseline changes with a vertical west-east trending strike-slip model crossing the network. We used a least square inversion to determine the slip rate of the fault which minimizes the differences between the observations and the strike-slip fault model[25]. This approach implicitly includes the assumption that baselines located on one side of the fault (i.e. not crossing the fault) are not changing. From baselines crossing the fault we found an optimal rate for strike-slip movement of 0.80 ± 1.25 mm yr$^{-1}$. This suggests that the surface fault slip rate across the network is close to zero and consistent with the results from the first six months of deployment[25]. Analysis of the geodetic data during the first six months of the deployment resulted in an upper bound on the slip rate of 6 mm yr$^{-1}$ only[25]. The seven-fold increase of fault slip resolution clearly demonstrates the need for long-term deployments in order to resolve tectonic processes.

**Modelling of strain and locking depth.** Next, we model strain for vertical strike-slip faulting. We use an analytical half-space solution, based on the elastic dislocation theory for an

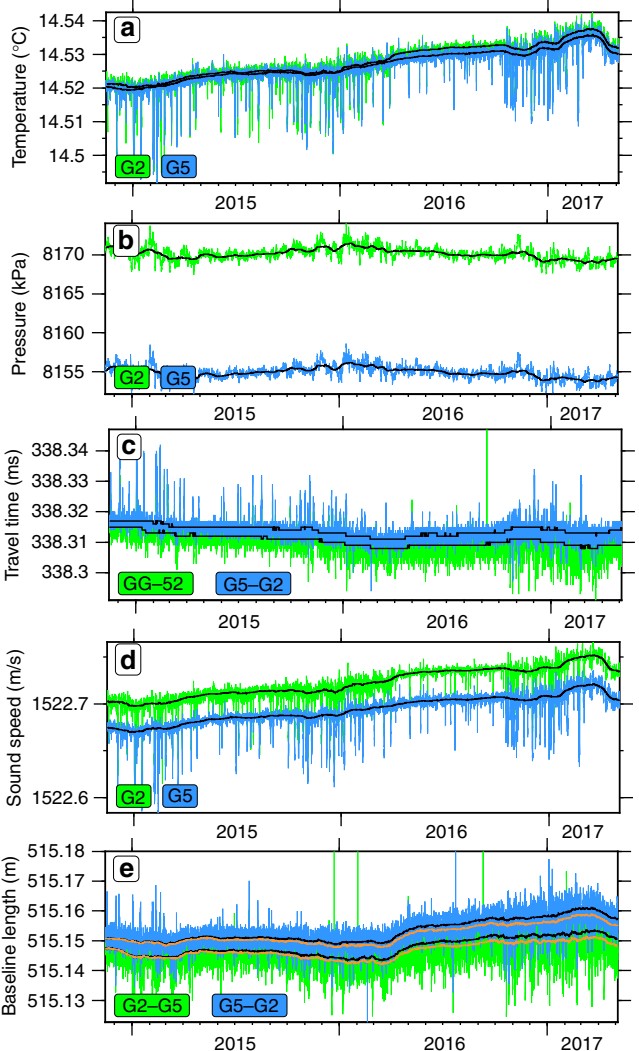

**Fig. 3** Measured parameters and estimated baseline for station pair G2-G5. Black lines indicate monthly medians. **a** Temperature. **b** Pressure. **c** Acoustic one-way travel times between the transponders for the forward- and backward measurement. **d** Estimated sound speed from temperature and pressure (using a constant salinity). **e** Resulting baselines for both measurement directions. Orange lines indicate the monthly mean baseline lengths with the inferred linear salinity decrease rate of 0.002 PSU yr$^{-1}$

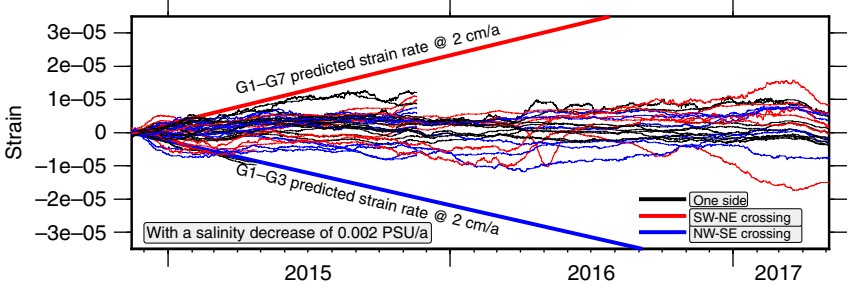

**Fig. 4** Strain of all baselines. Strain (monthly medians) estimated from the measured travel times and calculated sound speeds based on pressure, temperature and a linear salinity decrease of 0.002 PSU yr$^{-1}$. Baselines not crossing the fault (e.g. subparallel to the NAF) are shown in black; for a simple, east-west oriented, right-lateral strike-slip fault, baselines oriented southwest-northeast (red lines) should lengthen and those oriented northwest-southeast (blue lines) should shorten. Predicted strain rates correspond to a strike-slip movement reaching the seafloor

infinitely long vertical strike-slip fault[37]. Since the geodetic network is located on low rigidity sediments, we model the dependency of strain in the presence of a low rigidity layer (=weak layer). We use a simple model consisting of two horizontal layers with a rigidity contrast[37] to investigate the dependency of strain on creep below given depths, rigidity and varying thickness of the overlying sedimentary layer. The slip rates estimated for the NAF from onshore observations range between 15 and 27 mm yr$^{-1}$ [10,38,39] from GPS observations and between 15 and 19.7 mm yr$^{-1}$ from mass deposit considerations[40,41]. We model fault creep with a rate of 20 mm yr$^{-1}$ (in-between the GPS and geological estimates) for different faulting depths, above which the fault is locked and below which it creeps. The modelled strain is linearly dependent on the inferred slip rate of 20 mm yr$^{-1}$ [37]. We model scenarios for creep below 3 km (corresponding to the thickness of the basin sediments[36]) and hence strain in the pre-kinematic basement rocks (Fig. 5a) and for creep below 4.5 km depth (corresponding to the depth of the crystalline basement) (Fig. 5b). The depths of the geological units are known from a seismic profile passing in ~5 km distance south-east of the geodetic network[36]. The results show that the overlaying low rigidity sedimentary layers focus the strain close to the fault (Fig. 5a, b). Fault creep below 3 km (Fig. 5a), corresponding to slip in the pre-kinematic basement rocks[36], is clearly above the strain rate sensitivity of the geodetic network. The strain rate sensitivity of the geodetic network is $1.6 \times 10^{-6}$ yr$^{-1}$ corresponding to the inverted strike-slip movement of 0.8 mm yr$^{-1}$ considering the 500 m coverage of the geodetic network perpendicular to the fault (Fig. 5). Modeling the minimal fault slip inferred from onshore geodetic observations (16 mm yr$^{-1}$) results in 20% less strain and would still have been above to the sensitivity of the offshore geodetic network, in particular, due to the existence of weak shallow layers (Fig. 5a). From the modelling we find the locking depth of 3 km as the most conservative estimate. The strain rate induced from 20 mm yr$^{-1}$ creep in the crystalline basement (Fig. 5b), located below 4.5 km[36] depth, results in a signal exceeding the strain rate sensitivity for sedimentary layers thicker than 1 km.

Slip at depths below 16 km (Fig. 5c), corresponding to the inter-seismic deformation, results in a small strain signal and is still in line with the measurement uncertainty of strike-slip faulting of $0.80 \pm 1.25$ mm yr$^{-1}$. In the last step, we modelled the dependency of strain on rigidity contrasts between the upper and lower layer (Fig. 5d). We estimated the rigidity from empirical relations from a seismic P-velocity profile[36] (Supplementary Table 2). Despite the uncertainties of the empirical relations we find rigidity ratios clearly exceeding 10 for the material above and below the basement, so the calculated strain rate of strike-slip faulting on the seafloor is little dependent on these large values (Fig. 5d). From the strain rate estimates, we conclude that the strain rate sensitivity of the geodetic network is likely sufficient to resolve fault movement above the basement (Fig. 5a, b) and the model shows that the maximal resolution is reached for slip occurring at depths below 5.5 km (Fig. 5d).

With the possibility of distributed strike-slip across a few kilometre-wide zone of faults at the seafloor[26,30,42,43], we might not have captured the complete possible slip. However, deformation is clearly focussed beneath the geodetic network since the fault trace can be unambiguously identified in the bathymetry[26,30]. In particular, there is a clear 3.5 km right-lateral offset of a ridge between the baselines, indicating the location of the geodetic network above the main zone of surface deformation of the NAF[29].

**Local seismicity**. To better detect small-magnitude events indicative of a creeping behaviour, two small aperture OBS arrays were deployed in the vicinity of the geodetic stations and close to the NAF: a 5 km wide array during five months and a 12-km wide array for the next 12 months (Figs. 1 and 6). Such small aperture OBS arrays are significantly more sensitive to low-magnitude (from 0 and up) and shallow seismicity than larger aperture OBS arrays (e.g., 10 km station spacing) which have typically a magnitude of completeness of 1[13]. The OBS detections were complemented with phase picks from the land stations of the Kandilli Observatory and Earthquake Research Institute (KOERI). In 17 months, only 45 events with local magnitudes between 1.4 and 4.2 were detected and none are located near the geodetic experiment (Fig. 6). These events are 9 to 25 km deep and slightly offset (~2 km) from the NAF surface trace. The offset can be explained with the increased strain and hence stress accumulation in the vicinity of the locked fault[37] (Fig. 5c) and gas migrations at depth[42,44]. The local observation is in-line with the sparse seismicity recorded by onshore stations during the last decade along this segment of the NAF (Fig. 1) and, as the geodetic observation, also concurs with its quasi-locked status.

## Discussion

Seismicity below the Kumburgaz Basin is sparse (Figs. 1b and 6). This observation precludes a model where the fault would be locked at shallow depths but creeping at deeper levels (as modelled in Fig. 5), since seismicity would focus along the locking/creeping transition, as observed on the Prince Island segment[6] southeast of Istanbul (29.1°E). The seismic gap beneath the Kumburgaz Basin, together with the high level of seismicity on either side rather suggests that this section of the NAF is either completely locked or fully creeping. The observation of sparse seismicity, together with the absence of observable deformation from the geodetic network (Figs. 4 and 5) indicates a completely locked fault below the geodetic network down to larger depths.

Our results contrast with that of the Turkish-Japanese path-ranging[22] in the western part of the Sea of Marmara (red star in Fig. 1). Their data evidence a continuous dextral strike-slip deformation of $10.7 \pm 4.7$ mm yr$^{-1}$ in an area characterized by pronounced seismicity and where active creeping was previously inferred from repeating seismic events[11,12]. The seismicity patterns beneath both geodetic networks significantly differ but are consistent with the respective geodetic observations, where the fault segment with no seismicity seems locked, whereas the fault segment with a high level of seismicity is interpreted as being partially creeping. Although this observation is common for onshore faults, submarine faults may have a different behaviour due to the possible high amount of gas migrating upwards[42,44,45] and the large water content of the shallow sediments. The fault segment east of our geodetic network related to the Çınarcık Basin (CB) segment is characterized by seismicity which significantly increased[6] after the $M_w$ 7.4 1999 Izmit earthquake and was suggested to be related to stress increase imposed from the 1999 Izmit rupture zone to the eastern Sea of Marmara[6]. Although this segment is characterized by small magnitude seismicity, high-resolution hypocenter locations revealed that seismicity occurs almost exclusively below 10 km depth with very sparse seismicity at shallow depths[6], interpreted as being locked. Furthermore, seismicity in the eastern Sea of Marmara is characterized by the absence of repeating events suggested to be related to a locked fault[12].

Assuming a quasi-locked status of the NAF in the Sea of Marmara since its last known rupture in 1766, the accumulated slip deficit would be in the order of 4 m. We estimate (Supplementary Table 3) the equivalent moment magnitude with 7.1 magnitude units for an earthquake rupturing a 34 km-long fault segment of the NAF beneath the Kumburgaz Basin (Fig. 1). The

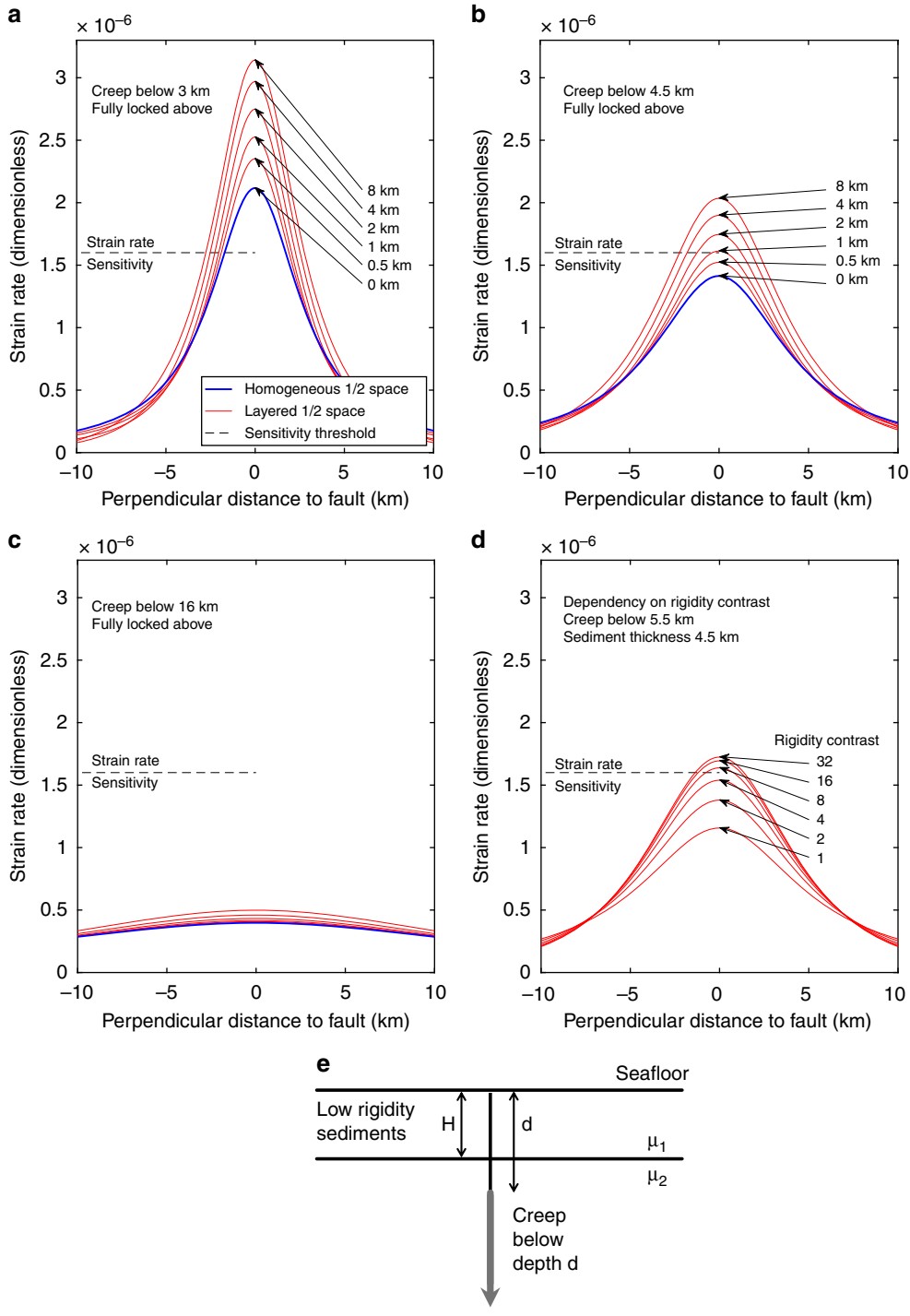

**Fig. 5** Modelled strain rates for a vertical strike-slip fault. The strain for homogenous and layered half space, based on elastic dislocation theory, was estimated using the analytical solution for a horizontally layered half-space and a vertical strike-slip fault[37]. **a** Model for creeping below 3 km depth corresponding to slip in the pre-kinematic basement rocks[36] and below. Above the locking depth, the fault is fully locked. Below the locking depth, the fault creeps at 20 mm yr$^{-1}$. The ratio of rigidity between the lower and upper layer is 4. The strain rate sensitivity of the geodetic network is 1.6*10$^{-6}$ yr$^{-1}$, corresponding to strike-slip movement of 0.8 mm yr$^{-1}$ considering a 500 m extension of the geodetic network perpendicular to the fault. **b** Model for slip below 4.5 km in the basement rocks[36]. **c** Creep below 16 km depth corresponding to interseismic deformation. **d** Dependency of rigidity contrast on the strain rate for a 4.5 km weak layer and creep below 5.5 km. Rigidity contrast from empirical relations is in the range between 8 and 16 (Supplementary Table 2). **e** Sketch showing the modelling setup and parameters. $\mu_1$ and $\mu_2$ are the rigidities of the shallow layer and below

slip deficit for a larger rupture involving all unbroken and locked segments in the Sea of Marmara Istanbul (e.g. rupture of the NAF in the Kumburgaz Basin and CB, Fig. 1) equals a moment magnitude 7.4 earthquake passing 25 km South of the city center of Istanbul. The magnitudes are in-line with previous studies[29,46]

and with the magnitudes of historical events along the NAF[1,3] and can be considered as a major hazard for the close-by Istanbul metropolitan area and its 15 million inhabitants.

Previous studies assumed a complete locking of the NAF in the eastern Sea of Marmara based on onshore observations since

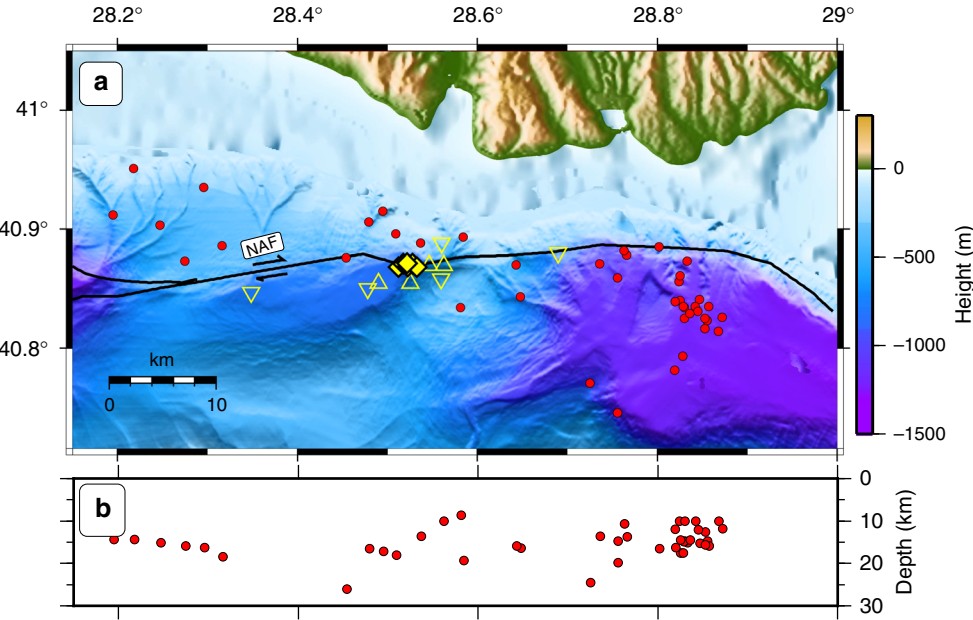

**Fig. 6** Seismicity located with the ocean bottom seismometer data. Microseismicity locations in map view (**a**) and projected along a vertical west-east trending profile (**b**). The OBSs were first deployed in a very small aperture array (29 October 2014 until 25 April 2015, upright triangles) and then re-deployed along the NAF (26 April 2015 until 13 April 2016; inverted triangles). Red circles show the events located with phases from OBS data combined with arrival times from land stations (KOERI). Yellow diamonds clustered near the OBSs show the geodetic stations. Bathymetry from ref. [30], topography from ref. [31] and fault traces from ref. [29]

in situ seafloor data were not available. Using observations from an intercommunicating network of acoustic transponders located on the seafloor and measuring across the NAF we show that the fault is locked down to at least 3 km and presumably down to 5.5 km depth, into the crystalline basement. Long-term OBS deployments designed to detect very small microseismicity reveal very sparse seismicity and absence of events directly beneath the OBS network. The geodetic monitoring together with the OBS observation indicate that the fault in the Kumburgaz Basin is fully locked. Together with recent results from a geodetic network[22] in the western Sea of Marmara, which revealed partial creep, our results indicate a complex fault locking pattern of the submerged NAF. Two unbroken and locked segments in the Sea of Marmara with accumulated strain equivalent to an earthquake between magnitude 7.1 and 7.4 remain and need to be considered in hazard assessments and risk estimates for the contiguous Istanbul metropolitan area. This study clearly demonstrates that in situ seafloor geodetic measurements along with OBS monitoring can fill observational gaps at sea and advocates the urgent need to conduct similar studies in regions with a high hazard potential from active faults offshore.

## Methods

**Direct path ranging method**. We measure the acoustic distance between two transponders by the two-way travel time of acoustic signals between the transponders and from the sound velocity, measured independently[22,23]:

$$s = v \cdot \frac{(\mathrm{TWT} - \mathrm{delta}_i)}{2} \tag{1}$$

where $s$ is the acoustic distance (i.e. baseline length), $v$ the sound velocity in water, and TWT the two-way travel time of the signal between the transponders and $\mathrm{delta}_i$ the response delay time of the responding station $i$ to the incoming signal. Due to the two-way travel time measurement, the acoustic signals travel in forward and backward direction and directional effects imposed by water fluxes cancel out. This is only possible since the temporal changes of water currents are clearly slower than our two-way travel times, which are mostly below 3 s. Overall little is known on the accurate water parameters on the seafloor of the Sea of Marmara because most stations measuring oceanic parameters are located at shallow water depths[47]. The temperature and pressure are measured at the active transponder during the TWT measurement; the velocity along the acoustic path is approximated by the

harmonic mean of the sound velocities of water at the stations (i.e. the endpoints of ray-paths).

**Acoustic transponders**. Ten acoustic distance metres, four from the Ocean Geosciences Laboratory in Brest (station names start with F) and six from Geomar's GeoSEA array in Kiel, Germany (station names start with G), were installed late October 2014[25] and fully operable until 5 May 2017. During the last visit with R/V Yunus on 29 January 2018 only three F-stations were running, all others stopped due to empty batteries. During the 2.5 years of complete operation of the network, each transponder measured temperature, pressure, inclination (Fig. 3 and Supplementary Figs. 1–3) and two-way travel times to the neighbouring stations (Supplementary Fig. 6).

**Sound velocity**. Each transponder had an integrated sound-velocity sensor but they showed unexpected offsets and long-term drift of up to 0.5 m s$^{-1}$ water speed in the time-series, which would map into apparent baseline changes of ~0.5 m. Although the sound speed measurement turned out useless for estimating the baseline lengths, they could be used to estimate salinity at the transponders in the order of 38.6 PSU, in-line with published data[48]. The sound speed (Fig. 3d, Supplementary Fig. 7) was calculated from temperature, pressure and the derived salinity value[32] assumed to have remained constant throughout the experiment.

**Salinity**. The sound speed $v$ can be calculated from pressure, temperature, and salinity[32]. We use the empirical relations, assuming a constant salinity, as in similar acoustic experiments[33,34].

**Temperature**. For the time after 5th May 2017 until January 2018 the F-stations were still operable, but the temperature sensors of the F-stations do not allow estimating a high-resolution baseline since artefacts of their temperature sensor cannot be isolated from a baseline change. Both networks operated independently but measured some common baselines (Fig. 2). We use the temperature data from the G-transponders for the close-by F transponders (distances less than 100 m each) since they have a high-resolution temperature sensor. The temperature time-series from the G-stations (sample interval of 90 min) were spline interpolated onto the measurement times of the close by F-stations (six measurements during the first 10 min of each hour).

Station F4 stopped sending out active baselines requests on 25 April 2015 but still responded to incoming baselines requests allowing to estimate the baselines for one-direction during the deployment (using pressure and temperature for F4 of close-by stations). The pressure sensor of transponder G2 was only working until 10 April 2016, later on, we used the pressure of G5 instead. Stations F2 (faulty temperature sensor) and G4 (temperature sensor broken after 19 November 2015) had both artefacts with the temperature sensors. Although close by to G1 the

temperatures of G1 could not be used for G4 nor F1 since both stations are likely located in a slightly different temperature field as suggested by the travel times measurements from G4 to all other transponders (Supplementary Fig. 6). Since the acoustic distance changes are dependent on sound speed variations (Eq. 1), which in turn are mostly influenced by temperature[32], the lack of accurate temperature measurements of G4 and F1 does not allow isolating a geologic movement using these stations. We extensively tried to use temperatures of G1 and G5 for G4 and F1, but this cannot compensate the different behaviour of travel time and due to the small heterogeneities of the temperature field and due to the trade-off between distance and water speed (Eq. 1) we did not use baselines from and to G4 and F1. Adopting the temperatures from other G-stations to G5 resulted in apparent baseline lengthening of ~1 cm to all other stations.

The temperature measurements (Fig. 3a and Supplementary Fig. 1) indicate a long-term increase of temperature with an approximate rate of 0.028 °C per year (~0.07 °C during the 2.5 years of deployment). Together with the temperature increase in the western part of the Sea of Marmara[22] (annual rate of 0.02 °C at the location indicated with a red star, Fig. 1), these observations might suggest general warming of sea bottom water in the Sea of Marmara. Furthermore, we observe repeated influx of cold water with an average temperature drop of 0.016 °C and average durations of two days. The temperature change can be tracked as the temperature successively drops within mostly ~10 h from east to west suggesting water mass movement of ~5 cm s$^{-1}$ and shows a seasonal effect with a higher rate of temperature drops in winter time (November-February).

**Pressure**. The pressure time series (Fig. 3b and Supplementary Fig. 2) for all sensors are very similar with variations similar to tide-gauge observation at the north coast of the Sea of Marmara (tide gauge station Marmara Ereğlisi, https://tudes.hgk.msb.gov.tr/tudesportal/, downloaded 2 September 2018). The pressure does not show a clear geological long-term trend as expected for a tectonic uplift or subsidence and we interpret the signal as long-term water changes above the transponders such as river-runoff, water in-flow from the Black Sea and the Dardanelles. For the baselines, pressure changes of 1 kPa would result in a baseline change of 11 mm for a 1000 m long baseline in case pressure change would not be compensated by travel times.

**Installation of geodetic frames on the seafloor**. First, potential sites along the NAF were mapped using high-resolution Autonomous Underwater Vehicle (AUV) bathymetry[26,27]. Then, 4 m high frames (Supplementary Fig. 8) with the transponder on top were lowered to the seafloor using a deep-sea cable. For accurate positioning on a target position, two transponders attached to the cable above the frame were used. Furthermore, buoyancy and weight were attached to the cable and above the frame, which allowed detecting the touch-down of the frames on the seafloor by observing the cable slack[49]. After their installation, all stations were inspected using a Remotely Operated Vehicle (Supplementary Figs. 8 and 9).

**Seismicity**. The OBSs were first deployed in a very small aperture array (29 October 2014 until 25 April 2015, upright green triangles) and then re-deployed along the NAF (26 April 2015 until 13 April 2016; inverted green triangles) (Fig. 1). The OBS network was installed in the shipping lane of the access to the Bosphorus in order to avoid station loss due to fishing activities. As a result, the elongated installation prevents very accurate estimates of the event depths using OBS only. Therefore, all events shown with red circles were located with land and OBS observations. Only events with RMS values smaller then 1 s and recorded on more than five stations are used. Events were located using the 1D velocity model from[50] and the seismological analysis software SEISAN[51]. The OBS stations did not significantly increase the number of events detected close to the geodetic network. Few additional events were detected compared to Koeri's catalogue, but P-S times (always larger than 2 s) and P polarisation indicate that these events did occur outside the OBS network and hence the geodetic network.

## Data availability
The geodetic data that support the findings of this study can be downloaded through the data repository PANGEA [G-stations, data https://doi.org/10.1594/PANGAEA.900275][52] and through SEANOE [F-stations, data https://doi.org/10.17882/59750][53]. The phase picks from the onshore seismological data is from KOERI (downloaded 15 June 2018). OBS data can be requested from the corresponding author.

## Code availability
The custom code used for the processing of the offshore geodetic data can be downloaded from https://github.com/flp-geo/geosea.

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

## Acknowledgements

The acoustic seafloor monitoring in the Sea of Marmara is a joint project of the Istanbul Technical University (Turkey) together with the University of Brest (France) and GEOMAR Helmholtz Centre for Ocean Research Kiel (Germany). We thank C. Hammersley and T. Bennetts from Sonardyne for their technical support with the geodetic instruments over the years. We thank the crews and Captains of R/V Pourquoi Pas?, R/V Poseidon, R/V Yunus and Oktay 8 for their support during installation and maintenance of the network. The GeoSEA Array is funded through grant 03F0658I of the BMBF (German Federal Ministry of Education and Research). Support was provided by the bilateral ANR/TÜBITAK collaborative research project MAREGAMI (ANR-16-CE03-0010-02 and Tübitak Project 116Y371). The French geodetic array was funded by the European Union, the Brittany Region and the French government (CPER ODO); support for the cruises was provided by CNRS-INSU through the European Monitoring Seafloor Observatory programme (EMSO).

## Author contributions

J.Y.R., V.B and H.K. funded and designed the experiment. D.L. and H.K. drafted the manuscript, which was revised and edited by all authors. P.H. implemented the analytical solution for the strain calculations, the modelling of strain was done by D.L. The GPS land-stations and calculation of the water column parameters were done by S.O., S.E. and F.P. The offshore geodetic dataset was independently processed witch different software written by D.L., F.P. who reached the same results. P.S. inverted the strike-slip rate from baseline data. The OBS data was processed by D.L. with input from L.G., J.B. and Z.C. All authors were involved in the configuration of the transponder network, network design, site selection, ship cruises, customs clearance, and logistics since the start of marine operations in 2014.

## Additional information

**Competing interests:** H.K., D.L. and F.P. are inventors of a patent to install stations on the seafloor using a deep-sea cable[49]. This method is not necessarily required for the installation of the stations which can also be done conventionally with a remotely operated vehicle or using a free-fall procedure. All other authors declare no competing interests.

