## [Peer Review File · Nature Communications]

Reviewers' comments:

Reviewer #1 (Remarks to the Author):

Review

Interseismic Strain Build-up on the Submarine North Anatolian Fault Offshore Istanbul
by D. Lange et al.

The manuscript deals with measurements of strain rates at the sea floor of the Sea of Marmara at a submarine segment of the North Anatolian Fault. This study is a valuable contribution to the discussion on the previously unclear state of the MMF, whether it is creeping or locked. The results presented give strong indication for a (nearly) complete locking of this fault segment that has not witnessed a strong earthquake for over 250 years, which is important knowledge for the assessment of seismic hazard.

The manuscript is well written and necessary background information is provided. I have no major objections regarding the manuscript but I would like to address a few points that concern the precision/reliability of the measurements and wish the authors to comment on these points.

1. Strain rates at the sea floor vs strain rates at seismogenic depth/in the basement. I assume that the sea floor where the sensors have been installed consists of soft sediments. Probably these sediments have very low elastic constants. This would mean that potential creeping motion on the fault at greater depth in the basement may be damped towards the sea floor in the soft sediments. Measured strain rates would then represent a lower bound of actual and relevant displacement rates at (seismogenic) depth. I would find it helpful to estimate this effect or comment on this issue by e.g. providing information on sediment thickness (variations) at the location of the array or estimated stiffness-depth profiles e.g. derived from shallow seismic velocities.

2. Vertical changes of sensors. I assume that the baseline length changes include not only the horizontal length changes but also potential vertical ones of the sensors. This could have the following implications. a) If the MMF is not vertical at the location of the array (the mentioned scarp at the sea floor is an indication for that) then vertical changes of the sea floor could interfere with the horizontal distances between the sensors. b) It is well known from leveling surveys onshore that thickness of soft sediments correlates with subsidence rates which is probably due to ongoing compaction of the sediments. I assume that sediment thickness increases towards the west of the sensor network in the Kumburgaz Basin and I would therefore expect higher subsidence rates at the western sensors compared to the eastern ones (besides tectonically-driven variations of vertical movements of the sea floor).

Probably, the vertical changes are negligible relative to the horizontal ones but I would like the authors to estimate or discuss whether their results could be affected by vertical changes.

3. Lines 192-194 Even if the fault is fully locked there should be some dextral shear deformation across the fault although the network's extent perpendicular to the fault is small. I recommend making use of analytical solutions to quantify expected dextral shear motion across the network in case of a fully locked fault (e.g. Savage & Burford 1973 or Okada 1985).

4. What is the reason for the different travel times, sound speeds and baseline lengths depending on in which direction measurements between two sensors are being made (e.g. Figs. 2 and 3)? I wonder if water currents play a role whose direction of flow have a component parallel to the direction of measurement. Is there anything known about water currents (except lines 426, 432, 433)? These could affect temperature, pressure and salinity. Canyons at the rims of the basins could indicate the existence of submarine currents. Water currents themselves would probably not bother the measurements (sound speed being much higher than potential water current speeds and measurement in both directions within short time intervals) unless they are non-steady state. In particular, I also note that the offset between measurements in different directions changes (increases) over time (see Figures 3 c,d,e). Is this due to drift of the sensors?

Small things

a) I recommend to provide a definition of the term “(this) (central) segment(s)” or to rephrase some passages as it appears to be used with different meanings (singular vs. plural; seismic gap vs. the linear fault segment between Central Basin and Çınarcık Basin; the red line in Figure 1 indicating the seismic gap is longer than the 70 km long segment referred to in line 231; see lines 27, 33, 66, 69, 70, 172, 187, 231).

b) Figure 3 b I don't understand why station G2 measures higher pressure than station G5 although water is deeper at station G5. Is bathymetry not resolved fine enough in Fig. 2?

c) Lines 148/149 c -> d ; d -> e

d) Line 160 green -> blue

e) Lines 172-174 This sentence also appears in lines 214-217

f) Lines 224/225 insert “the” two times and Basin capital B: “... western edge of the Çınarcık Basin ... beneath the Kumburgaz Basin ...”

g) Line 232 GPa

h) Line 374 two times “velocities”

i) Line 438 should be Fig. 1

j) Line 442 then -> than

k) Lines 449/450 two times “OBS data can be downloaded at :[to be provided]”

Supplementary Material

l) Supplementary Figure S1. It seems that the temperature slightly drops or becomes more or less constant towards the end of a year before rising again in a new year. Could this be a seasonal effect (e.g. convection currents due to seasonal changes of water temperature at different depths)? Or is it because the number of “events (??)” (Line 423 main text) increases towards the end of a year, with fewer events in summer?

m) Supplementary Figure S1b. It would be good to use colour lines for the average as in S2b instead of black lines for all stations to be able to discriminate between them.

n) Supplementary Figure S2b. Stations F2 and G5 are installed nearby each other, nevertheless station F2 measured a relative increase in pressure compared to station G5. This could mean that there are occurring pressure changes within small distances or the measurement devices show some kind of drift or different systematic measurement errors.

o) Caption of Supplementary Figure S3. “a 4 m high structure”: is this the height of the foundations of the stations? How are the stations installed, are they rammed into the sea floor?

p) Line 57 Erase “the”

q) Line 60 insert “in”: discussed in the text

r) Line 61 insert “of”: ...using a strain rate of 1.8-6 per year

s) Line 63 discussion instead of discussing

t) Supplementary Figure S5. Are the different depths of the stations accounted for in order to get horizontal baseline length changes or are the vertical components of length changes included (they are probably negligible relative to the horizontal ones)?

u) Line 78 baselines instead of baseline

T. Hergert

Reviewer #2 (Remarks to the Author):

Measurement result and scientific interpretation seem reasonably sound. Writing is ok, but lacks some sharpness, particularly when reporting the numerical results and its statistical significance. Specifically, lines 188-198 are unclear. Displacement is reported as 2 mm +/- 3.1 mm over the 2.5 years span. Thus, wouldn't the average rate would be 0.8 mm/a? If you are making a different point it needs to be explained more clearly.

Text at lines #214-217 is almost exactly the same as 172-274.

Reference #9 and #26 the same. #35 is only a partial reference.

Some suggested rewording for this text and other is included on the attached annotated manuscript.

Reviewer #3 (Remarks to the Author):

Interseismic Strain Build-up on the Submarine North Anatolian Fault Offshore Istanbul
By D. Lange, H. Kopp, J.-Y. Royer, ..., L. Géli

This manuscript investigated seafloor geodetic observation beneath the Sea of Marmara and demonstrates no significant creep detected across the North Anatolian Fault, hence fully locked. It should be very important result to assess strain accumulation for a large earthquake in the future. The observation and analysis seem to be appropriate. However, it may be much a summary of technical reports. Moreover, although Nature Communications requires novelty results, this result seems to be not much different from previous one (e.g., Sakic et al., 2016). Therefore, I would suggest that this paper cannot be acceptable unless the novelty is clarified and following issues are revised.

First of all, I would suggest following major points.

- 1) As mentioned above, please clarify the novelty in this research comparing with previous ones (e.g., Sakic et al., 2016). It may be necessary to drastic improvement of analysis method, and/or novel discussion (e.g., 2-D fault slip simulation). Although it is certainly interesting about ocean bottom seismographs, Yamamoto et al. (2017) had already discussed.
- 2) It would be helpful to include why the authors discuss strain instead of slip rate. In the introduction section, they described importance that “(i) fully locked or (ii) creeping (Line 49)”. Nevertheless, they converted slip amount into strain (Line 128–).
- 3) Please include enough information that how were along-fault baselines treated. It is difficult to understand because of converting into strain rate, but the baseline along fault seems to be shortening (e.g., G7–G3) or extension (G1–G2).
- 4) For all baseline changes, slip rate (e.g., linear regression) and their error should be specified in the analysis part. It would prefer to show the table in supplementary material.
- 5) The method of integration for all baseline changes is unclear. Please clarify the authors used only baselines across-fault, or also used baselines along-fault. Do they calculate a simple average when integrating all baselines?
- 6) Comments of the Istanbul earthquake hazard should be included in the discussion section. As ‘Istanbul’ is written in the title, readers would want to know the relationship between this study and it.

There are some minor revisions.

- 7) While the authors present how to analyze (Line 110–141), it should not be ‘Discussion’ section. Analysis and discussion should be separated into two sections.
- 8) Fig.2: It would prefer to use the unit ‘mm/a’ instead of ‘mm’.
- 9) More explanation on pressure sensor’s drift should be given in Line 139–141. The authors should

provide a reference why “uplift is unlikely”.

10) ‘Marmara Sea’ should be replaced by ‘Sea of Marmara’ throughout the manuscript (e.g., Line 28).

I hope these comments will be helpful.

Dear Reviewers,

In this document, we address your comments and helpful suggestions. Please find our answers below marked in **blue** (original comments marked in **black**).

We start by addressing the comments about the novelty of our work in light of three recent publications.

We have taken special care to emphasize the novelty of our work in the revised manuscript, which we understand was not clearly exposed in the original version. In addition to our answer here also refer to our answer to Comment 1) raised by Reviewer #3 below.

Here we summarize the advance of our work compared to the three previous papers:

a.) The *Sakic et al.* (2016) paper was a methodological study, introducing methodical aspects of strike-slip inversion of offshore geodetic data and the resulting uncertainties therefrom. As a methodological paper, the authors did not take any additional data or modeling into account and could not evaluate the geodetic data in conjunction with seismicity or fault geometry. Furthermore, they did not assess the inversion results to a model of deformation along a strike-slip fault, as the limited observation period of only 6 months returned an insufficient resolution of the deformation rate to do so. Thus, due to the limited resolution, *Sakic et al.* could not evaluate the results of their method against an elastic dislocation theory, which takes e.g. rigidity, locking depth or distributed deformation away from the main fault trace into account. **In contrast, our current manuscript, while partially relying on the same method, is focused on the tectonic analysis and numerical modeling of slip along a fault in a multidisciplinary approach (including results from two dedicated OBS networks and numerical modeling of varying rigidity contrasts, locking depth, fault geometries etc.). This is the crucial advance compared to the *Sakic et al.* paper and is only possible because of the extended observation period (2.5 yrs) which yielded a seven-fold increase in resolution of the fault slip.** Furthermore, the original 6 months of data used by *Sakic et al.* was too limited to exclude the possibility of temporal changes in faulting behavior (i.e. changes in fault locking or episodic slip events), which the increased resolution of our current work is capable to resolve. Finally, our current work discusses the geodetic results in the regional setting of the Sea of Marmara, whereas *Sakic et al.*, being a methodological paper, only looked at very local effects.

b.) The OBS study from *Yamamoto et al.* (2017) is an important contribution on the seismic segmentation of the NAF in the Sea of Marmara, revealing clusters of enhanced seismic activity vs regions of low seismicity. Their OBS deployment was optimized to capture seismicity throughout the Sea of Marmara (10+5 OBS distributed over a distance of approx. 100 km lateral extent, resulting in a lateral station spacing of approximately 12 km, see their Figure 1). In contrast, all our stations were installed very close (<5 km) to the NAF around the geodetic network with an aperture of ~5 km of the array. The main target of our OBS deployment was to detect extremely small events to assure that the fault is surely not moving beneath the geodetic network and its direct vicinity. To this end, the two OBS deployments (*Yamamoto et al.*, 2017 and this study) are targeting completely different scales (5 km vs 100 km) to address different, but related scientific hypothesis: *Yamamoto et al.* aim to verify the hypothesis that the NAF shows heterogeneous, segmented locking based on the OBS monitoring of lateral changes of hypocenter distributions across the entire Sea of Marmara. In contrast, our very local, dedicated OBS deployment aims to verify the hypothesis that the absence of significant slip along the NAF is accompanied by the absence of seismicity at that specific location.

The scientific advance of our study compared to the earlier paper of *Yamamoto et al.* (2017) is

- 1) the combination of seafloor geodesy & OBS monitoring and
- 2) and more importantly, our approach lets us distinguish between a fault behavior characterized by full creep vs full locking, which is a crucial prerequisite for strain build-up and consequently for the hazard assessment of Istanbul.

c.) We were very pleased to see the study of *Yamamoto et al. (2019)* published last year because their results are highly relevant to our manuscript. *Yamamoto et al.* observed creep in the western part of the Sea of Marmara, consistent with well-known small repeating earthquake event activity. In contrast, our results indicate no significant slip in the central part of the Sea of Marmara, which is inferred to be locked due to the absence of seismicity. The main scientific advance from our study with regards to the *Yamamoto et al. (2019)* study is the validation that the physical state (creeping vs locking) of the NAF is highly heterogeneous in the Sea of Marmara. While the varying seismicity pattern along the offshore portion of the NAF was known from previous studies considering only earthquake data, these studies could not distinguish between fault creeping and fault locking. **The significance of our study results is increased through the findings of *Yamamoto et al (2019)* as it is now clear that in some segments of the NAF the fault is locked (our findings) whereas other segments are creeping (*Yamamoto et al., 2019*).** Without the *Yamamoto et al.* paper the creeping case would not have been documented. **With our findings now in conjunction with *Yamamoto et al. (2019)* the NAF is the only offshore fault for which these contrasting styles are actually measured and monitored.** Although this observation is common for onshore faults, submarine faults may have a different behaviour due to the high amount of gas migrating upwards and the large water content of the shallow sediments. **Finally, our findings bear a strong impact on the hazard assessment for the Istanbul metropolitan area due to 1) the proximity of our network to the city, whereas the network of *Yamamoto et al. (2019)* is located at the western termination of the Sea of Marmara and 2) our measurements did not detect any significant slip (in contrast to the *Yamamoto et al.* network) indicating strain build-up along this segment of the NAF (again in contrast to the western segment where the *Yamamoto et al.* array was located).**

We further ask you to strengthen your dataset by the application of analytical methods, e.g. numerically describing the coupling rate (as done in the Sakic 2016 study).

The most significant and comprehensive change to the original version of our manuscript is the addition of a numerical analysis and parameter study on the dependency of surface strain and locking depth for a strike-slip fault considering the influence of low rigidity layers, not covered in previous studies in the region. This led to a major and substantial revision of the manuscript, where the new analysis of the data now occupies the majority of text in the Results and Discussion section. These changes have in our view tremendously improved the manuscript and address the criticisms and concerns of the reviewers.

Finally, to broaden the interest of your study, we would ask you to discuss the seismic (and potentially tsunami) hazard around Istanbul, based on your results.

Regarding the seismic and tsunami hazard we start now the introduction with:

“It is well known that Istanbul city and populations along the coasts of the Sea of Marmara were previously severely affected by earthquakes related to the submerged North Anatolian Fault (NAF) in the Sea of Marmara¹. Some of the earthquakes were associated with seismically driven sea-waves and six destructive run-ups are known from historical reports for the last 20 centuries². For example, the 1766 earthquake, suggested to have nucleated beneath the western Sea of Marmara³, resulted in very strong shaking in Istanbul (Mercalli Intensity VII, “very strong” shaking) and seismically driven sea-waves submerged the quays in Istanbul². “

Seismic hazard is again discussed in the Discussion and Conclusions section of the manuscript, where we now discuss seismotectonic segmentation and quantify the potential earthquake magnitude of a rupture along this segment of the NAF.

Reviewers' comments:

Reviewer #1 (Remarks to the Author):

Review

Interseismic Strain Build-up on the Submarine North Anatolian Fault Offshore Istanbul
by D. Lange et al.

The manuscript deals with measurements of strain rates at the sea floor of the Sea of Marmara at a submarine segment of the North Anatolian Fault. This study is a valuable contribution to the discussion on the previously unclear state of the MMF, whether it is creeping or locked. The results presented give strong indication for a (nearly) complete locking of this fault segment that has not witnessed a strong earthquake for over 250 years, which is important knowledge for the assessment of seismic hazard.

The manuscript is well written and necessary background information is provided. I have no major objections regarding the manuscript but I would like to address a few points that concern the precision/reliability of the measurements and wish the authors to comment on these points.

1. Strain rates at the sea floor vs strain rates at seismogenic depth/in the basement. I assume that the sea floor where the sensors have been installed consists of soft sediments. Probably these sediments have very low elastic constants. This would mean that potential creeping motion on the fault at greater depth in the basement may be damped towards the sea floor in the soft sediments. Measured strain rates would then represent a lower bound of actual and relevant displacement rates at (seismogenic) depth. I would find it helpful to estimate this effect or comment on this issue by e.g. providing information on sediment thickness (variations) at the location of the array or estimated stiffness-depth profiles e.g. derived from shallow seismic velocities.

We totally agree on the effect of weak sediments on the displacements. However, the strain is not damped due to the existence of low rigidity sediments close to the surface. In fact, a weak (e.g. lower rigidity) upper layer enhances strain close to the fault and reduces strain away of the fault (Rybicky, 1971, BSSA, Ma & Kuszniir, 1994, PAGEOPH), which is somewhat counterintuitive, so we explored this further for our case along the NAF. We included a new Figure 5 showing the dependency of horizontal strain on slip below different depths and varying overlying layers of low rigidity. Strain close to the fault (distances smaller than ~2.5 km) is focused, and strain accumulation close to the fault strongly increases even in case a moving fault extends into the overlaying low rigidity sediment layer. The basement depth in the region of the central high was imaged at depths of ~4.5 km (Bécel et al., 2010, their Figure 10) from active seismic profiling, located in close proximity to our study site (~5 km). Using Bécel's v_p velocity profile we estimated rigidity (=stiffness) using classical empirical formulas (Gardner's relation and mudrock line). We added a new table with the estimated rigidity-depth profiles in the supplementary material (supplementary Table 2). Figure 5d shows that the dependency of strain in presence of significant rigidity contrasts (as the rigidity estimate suggests) is small.

In general, sedimentary layers in the Sea of Marmara are inclined in the same direction as the bathymetry (e.g. Bécel et al., 2010) due to the formation of the subsiding basin. The inclination for the uppermost 2 km of sediments mimics the inclination of the bathymetry. The best information on sediment thickness variations is available for a profile located ~5 km

south-west of the geodetic deployment (Bécel et al. 2010, Figure 5). From the transition from the Central High to the Kumburgaz basin, inclination of sediments is similar to the bathymetry inclination and is $\sim 5.7^\circ$. Since the inclination of bathymetry beneath the geodetic network is less than 3° (see new Table S2) we did not add a discussion of the inclination of basin sediments.

2. Vertical changes of sensors. I assume that the baseline length changes include not only the horizontal length changes but also potential vertical ones of the sensors. This could have the following implications. a) If the MMF is not vertical at the location of the array (the mentioned scarp at the sea floor is an indication for that) then vertical changes of the sea floor could interfere with the horizontal distances between the sensors. b) It is well known from leveling surveys onshore that thickness of soft sediments correlates with subsidence rates which are probably due to ongoing compaction of the sediments. I assume that sediment thickness increases towards the west of the sensor network in the Kumburgaz Basin and I would therefore expect higher subsidence rates at the western sensors compared to the eastern ones (besides tectonically-driven variations of vertical movements of the sea floor).

Probably, the vertical changes are negligible relative to the horizontal ones but I would like the authors to estimate or discuss whether their results could be affected by vertical changes.

We added the inclinations of the baselines to Supplementary Table 1. Apart from station G3, located on the slope to the south-east, all baselines are inclined by less than 1.3° and always smaller than 3° .

We agree that the effect of vertical displacements mapping onto horizontal distances is well known from geodetic measurements. However, it is known as well that this effect only plays a role for larger angles. Since we now added the angles in the supplementary, we did not change the manuscript further. Similarly, we did not include a discussion on possible different vertical settlements in the small-scale network from different sediment thicknesses.

To document that the horizontal distances are not influenced by vertical movement we provide details and a figure:

Height change of one instrument changes the baseline length:

with

b = baseline length, Δh = vertical change and Δl = length change of baseline.

Results to

$$\sin \alpha = \frac{\Delta h}{b + \Delta l} \Rightarrow \Delta l = \frac{\Delta h}{\sin \alpha} - b$$

Next, insert

$$\tan \alpha = \Delta h / b$$

in the equation above to get $\Delta l = f(\Delta h, b)$:

$$\Delta l = \frac{\Delta h}{\sin \operatorname{atan} \frac{\Delta h}{b}} - b$$

In Fig. R1 we plotted the dependency of vertical movement on horizontal baselines for baseline lengths of 500, 800 and 1700 m. Vertical changes of a station of at least 1.75 m would be needed to change a baseline more than 3 mm. Vertical movement exceeding some centimeters would have been surely registered by the pressure sensors. Please note, that the formula above is only 100% accurate for horizontal baselines. However, for small angles (e.g. $b \gg \Delta h$) Fig. R1 would be nearly identical because for small angles θ trigonometric approximations can be used ($\sin \theta \approx \theta$ and $\tan \theta \approx \theta$).

Fig R1. Dependency of baseline change on vertical displacement. The three lines indicate dependency for 500 m, 800 m and 1700 m long baselines, respectively.

3. Lines 192-194 Even if the fault is fully locked there should be some dextral shear deformation across the fault although the network's extent perpendicular to the fault is small. I recommend making use of analytical solutions to quantify expected dextral shear motion across the network in case of a fully locked fault (e.g. Savage & Burford 1973 or Okada 1985).

We have investigated this effect in the new numerical modeling and parameter study. See Figure 5c.

4. What is the reason for the different travel times, sound speeds and baseline lengths depending on in which direction measurements between two sensors are being made (e.g. Figs. 2 and 3)? I wonder if water currents play a role whose direction of flow have a component parallel to the direction of measurement. Is there anything known about water currents (except lines 426, 432, 433)? These could affect temperature, pressure and salinity. Canyons at the rims of the basins could indicate the existence of submarine currents. Water currents themselves would probably not bother the measurements (sound speed being much higher than potential water current speeds and measurement in both directions within short time intervals) unless they are non-steady state.

We could not find a dependency of baseline length on the azimuthal baseline direction, travel times and baseline lengths (Figs. 2 and 3). We added the following text and one reference (Aydoğdu et al., 2018) about water circulation of the Sea of Marmara to the method section:

“Due to the two-way travel time measurement, the acoustic signals travel in forward and backward direction and directional effects imposed by water fluxes cancel out. This is only possible since the temporal changes of water currents are clearly slower than our two-way travel times, which are mostly below 3 s. Overall little is known on the accurate water parameters on the seafloor of the Sea of Marmara because most stations measuring oceanic parameters are located at shallow water depths⁴⁴. The temperature and pressure are measured at the active transponder during the TWT measurement; the velocity along the acoustic path is approximated by the harmonic mean of the sound velocities of water at the stations (i.e. the endpoints of ray-paths).”

In particular, I also note that the offset between measurements in different directions changes (increases) over time (see Figures 3 c,d,e). Is this due to drift of the sensors?

Again, we cannot find directional dependency of measurements. Please note that pressure (Fig. 3b) is based on measurements at the stations (e.g. point measurements, omnidirectional). Similarly, sound speed (Fig. 3d) is based on pressure and temperature (e.g. point measurements) and should, therefore, not depend on the direction.

We added this sentence:

*“The data suggest an absence of vertical movement and the remaining small baseline changes originate from measurement uncertainties of the pressure and temperature sensors. The subtle changes of water parameters such as the temperature increase of 0.002°C on March 2016 (Fig. 3a) might be a sensor artefact since they are not compensated by pressure or travel time resulting in an apparent ~3 mm baseline length change (Fig. 4d). However, since baseline estimates are based on the equation $distance=time*velocity$, travel times and water sound-speeds must be jointly accurately known. This problem is similar to the hypocenter-depth-velocity dependency in earthquake location techniques. This is the reason why the network was designed to measure a high number of baselines across the fault to allow isolating effects of sensor drift from baseline changes.”*

a) I recommend to provide a definition of the term “(this) (central) segment(s)” or to rephrase some passages as it appears to be used with different meanings (singular vs. plural; seismic

gap vs. the linear fault segment between Central Basin and Çınarcık Basin; the red line in Figure 1 indicating the seismic gap is longer than the 70 km long segment referred to in line 231; see lines 27, 33, 66, 69, 70, 172, 187, 231).

We deleted the red line and added the lateral extent of sedimentary basins in the profile of Figure 1. We modified different paragraphs of the text about the fault segmentation. Supplementary Table S3 lists the fault lengths and seismic moments of the remaining locked segments.

b) Figure 3 b I don't understand why station G2 measures higher pressure than station G5 although water is deeper at station G5. Is bathymetry not resolved fine enough in Fig. 2?

This is a good point. Indeed, it turned out we did not use the latest AUV bathymetry and updated Fig. 2 accordingly. The pressure of instrument G2 indeed indicates a horizontal position 1.53 m deeper than G5 and is now in-line with the updated AUV bathymetry shown in Figure 2. The difference between the AUV bathymetry and the pressure data is now less than 1.5 m for all stations. We also carefully re-checked all horizontal station coordinates from the installation procedure and the measured distances are all in-line with the station positions.

c) Lines 148/149 c -> d ; d -> e
Done.

d) Line 160 green -> blue
Done.

e) Lines 172-174 This sentence also appears in lines 214-217
Right, we deleted the second occurrence (Line 214-217).

f) Lines 224/225 insert "the" two times and Basin capital B: "... western edge of the Çınarcık Basin ... beneath the Kumburgaz Basin ..."
Done.

g) Line 232 Gpa
Done.

h) Line 374 two times "velocities"
Done.

i) Line 438 should be Fig. 1
Done.

j) Line 442 then -> than
Done.

k) Lines 449/450 two times "OBS data can be downloaded at :[to be provided]"
We currently discuss the data access with our data management team in Kiel/Germany and will be able to provide persistent links to the data soon. The F-geodetic stations are already accessible using a DOI.

Supplementary Material

l) Supplementary Figure S1. It seems that the temperature slightly drops or becomes more or less constant towards the end of a year before rising again in a new year. Could this be a seasonal effect (e.g. convection currents due to seasonal changes of water temperature at

different depths)? Or is it because the number of “events (??)” (Line 423 main text) increases towards the end of a year, with fewer events in summer?

The physical mechanisms of the temperature drops are not well understood and seem only to occur on the seafloor since they are not known from other water temperature measurements, which are mostly close to the coast and harbors and in shallow water. We think that there might be a small increase of the warming period after winter time, but the modelling or statistical significance of this subtle increase and the underwater water currents of the Marmara Sea are not the scope of this paper. However, our Turkish colleagues are currently integrating the temperature data in their oceanographic model (Prof. Emin Özsoy, ITU).

We added this sentence to the Method/Temperature section (Line 675):

“... suggesting water mass movement of ~5cm/s and shows a seasonal effect with a higher rate of temperature drops in winter time (November-February).”

Since the paper from Yamamoto et al., 2019 observed a similar temperature increase we added to line 564:

“... Together with the temperature increase in the western part of the Marmara²² (annual rate of 0.02°C at the location indicated with a red star, Figure 1), these observations might suggest general warming of sea bottom water in the Sea of Marmara.”

m) Supplementary Figure S1b. It would be good to use colour lines for the average as in S2b instead of black lines for all stations to be able to discriminate between them.

We changed the black lines in S1b to colour lines as suggested.

n) Supplementary Figure S2b. Stations F2 and G5 are installed nearby each other, nevertheless station F2 measured a relative increase in pressure compared to station G5. This could mean that there are occurring pressure changes within small distances or the measurement devices show some kind of drift or different systematic measurement errors.

We added the following text passage:

“F2 measured a relative increase of approximately 1 kPa (equivalent to 10 cm water column change) relative to closeby station G5 which we interpret as most likely due to a systematic pressure drift of station F2. In general, the pressure sensors used are known to have a long term mean drift of 0.88+/-0.73 kPa/a¹¹. As a result, drift and the differences between the pressure measurements might all be explained with sensor drift. Resolution of pressure is around 10 Pa corresponding to an effective depth resolution of ~1 mm.”

o) Caption of Supplementary Figure S3. “a 4 m high structure”: is this the height of the foundations of the stations? How are the stations installed, are they rammed into the sea floor?

We inserted in the “Methods” section a paragraph and a reference about the installation procedure:

“Installation of geodetic frames on the seafloor. First, potential sites along the NAF were mapped using high-resolution Autonomous Underwater Vehicle (AUV) bathymetry^{29,30}. Then, 4 m high frames (Supplementary Fig. 8) with the transponder on top were lowered to the seafloor using a deep-sea cable. For accurate positioning on a target position, two transponders attached to the cable above the frame were used. Furthermore, buoyancy and weight were attached to the cable and above the frame, which allowed detecting the touch-

down of the frames on the seafloor by observing the cable slack⁴⁶. After their installation, all stations were inspected using a Remotely Operated Vehicle (ROV) (Supplementary Fig. 9)."

We added Fig. S8 and S9 to the supplementary material showing the frames on the seafloor and an image of the seafloor.

p) Line 57 Erase "the"

Done.

q) Line 60 insert "in": discussed in the text

Done.

r) Line 61 insert "of": ...using a strain rate of $1.8 \cdot 10^{-6}$ per year

Done.

s) Line 63 discussion instead of discussing

Done.

t) Supplementary Figure S5. Are the different depths of the stations accounted for in order to get horizontal baseline length changes or are the vertical components of length changes included (they are probably negligible relative to the horizontal ones)?

Totally right. To order to clarify this we added to the caption of Figure S5:

"The baselines are measured in the direction of the curved acoustic ray² traveling from one transponder to the other and therefore include the horizontal and vertical components of the length changes."

See as well point #2 where we estimate the dependency of the horizontal length changes on vertical movement.

u) Line 78 baselines instead of baseline

Done.

T. Hergert

Reviewer #2 (Remarks to the Author):

Measurement result and scientific interpretation seem reasonably sound. Writing is ok, but lacks some sharpness, particularly when reporting the numerical results and its statistical significance.

Specifically, lines 188-198 are unclear. Displacement is reported as 2 mm +/- 3.1 mm over the 2.5 years span. Thus, wouldn't the average rate would be 0.8 mm/a? If you are making a different point it needs to be explained more clearly.

Right, the average slip rate is 0.8 mm/a. Following this suggestion and Reviewer #3 (point 3,5) we changed Figure 2 to slip rates, and now discuss slip rates.

We re-phrased this paragraph accordingly:

"We compared the observed baseline changes with a vertical west-east trending strike-slip model crossing the network. We used a least square inversion to determine the slip rate of the fault which minimizes the differences between the observations and the strike-slip fault model²⁸. This approach implicitly includes the assumption that baselines located on one side of the fault (i.e. not crossing the fault) are not changing. From baselines crossing the fault we found an optimal rate for strike-slip movement of 0.80 ± 1.25 mm/a. This suggests that the surface fault slip rate across the network is close to zero and consistent with the results from

*the first six months of deployment*²⁸. Analysis of the geodetic data during the first six months of the deployment resulted in an upper bound on the slip rate of 6 mm/a only²⁸. The seven-fold increase of fault slip resolution clearly demonstrates the need for long-term deployments in order to resolve tectonic process.“

Text at lines #214-217 is almost exactly the same as 172-274.
Corrected, we deleted line #214-217.

Reference #9 and #26 the same. #35 is only a partial reference.
Corrected.

Some suggested rewording for this text and other is included on the attached annotated manuscript.

We included the suggestions from the annotated PDF in the manuscript.

Reviewer #3 (Remarks to the Author):

Interseismic Strain Build-up on the Submarine North Anatolian Fault Offshore Istanbul
By D. Lange, H. Kopp, J.-Y. Royer, ..., L. Géli

This manuscript investigated seafloor geodetic observation beneath the Sea of Marmara and demonstrates no significant creep detected across the North Anatolian Fault, hence fully locked. It should be very important result to assess strain accumulation for a large earthquake in the future. The observation and analysis seem to be appropriate. However, it may be much a summary of technical reports. Moreover, although Nature Communications requires novelty results, this result seems to be not much different from previous one (e.g., Sakic et al., 2016). Therefore, I would suggest that this paper cannot be acceptable unless the novelty is clarified and following issues are revised.

First of all, I would suggest following major points.

1) As mentioned above, please clarify the novelty in this research comparing with previous ones (e.g., Sakic et al., 2016). It may be necessary to drastic improvement of analysis method, and/or novel discussion (e.g., 2-D fault slip simulation). Although it is certainly interesting about ocean bottom seismographs, Yamamoto et al. (2017) had already discussed.

Please also see the text to the editor's concerns about the novelty of this study at the top of this document.

During the installation of our OBS network in November 2014, we were aware of the OBS network from Yamamoto et al. (2017). We still decided on another installation focussing on the geodetic network because:

- a.) Our OBS stations are very close to the fault trace and we installed 4 OBS stations along 6 km to have the highest detection threshold for microseismicity (29 October 2014 until 25 April 2015, upright green triangles, Figure 5).
- b.) Later on, between April 2015 and April 2016 we were running 5 OBS stations along the NAF, therefrom four stations with less than 6 km distance from the geodetic array (inverted green triangles, Figure 5).

In comparison, the Yamamoto et al. (2017) network consists of three stations on the central high (our working area), therefrom two stations ~5 km away from the main trace and only one on the NAF (Stations 11,12,13, Figure 1, Yamamoto et al., 2017). Installation time for Yamamoto et al.'s network was between September 2014 and July 2015 only.

Overall, our deployments were completely optimized for the best detection of local seismicity, while Yamamoto et al. (2017) optimized on seismic events in the whole Marmara Sea due to their much larger station spacing of 10 km. To clarify we added:

“To better detect small-magnitude events indicative of a creeping behaviour, two small aperture OBS arrays were deployed in the vicinity of the geodetic stations and close to the NAF: a 5 km wide array during five months and a 12-km wide array for the next 12 months (Figs. 1 and 6). Such small aperture OBS arrays are significantly more sensitive to low-magnitude (from 0 and up) and shallow seismicity than larger aperture OBS arrays (e.g. 10 km station spacing) which have typically a magnitude of completeness of 1^{13} ”.

Please see the additional comments to the editor’s concerns at the top of this document, discussing the novelty of our study in relation to Sakic et al, 2016 and Yamamoto et al, 2017, 2019.

2) It would be helpful to include why the authors discuss strain instead of slip rate. In the introduction section, they described importance that “(i) fully¹⁻³ locked or (ii) creeping (Line 49)”. Nevertheless, they converted slip amount into strain (Line 128–).

From the travel time, we measure distance changes in-between stations which we convert into strain ($\Delta L/L$) since potential influence of the water parameters map linearly into distance and therefore strain. The strain indicates the deformation on the seafloor. Then, we convert the observed strain (measured across the fault) to slip on a W-E trending strike-slip fault. Throughout the manuscript, we use “*strain*” and fault “*slip*” always as different and distinct words. In (previous) Line 128- we only discuss strain and the term slip is not used in this paragraph nor the following. Since we think it’s a wording issue with strain and slip we modified various occurrences of the word “*slip*” to words like “*fault slip*” or “*strike-slip displacement*”.

3) Please include enough information that how were along-fault baselines treated. It is difficult to understand because of converting into strain rate, but the baseline along fault seems to be shortening (e.g., G7–G3) or extension (G1–G2).

We extended the section (Lines 191-202) on the estimation of the optimal strike-slip of the fault. Please see as well the first point of reviewer #1.

There we state:

“...This approach implicitly includes the assumption that baselines located on one side of the fault (i.e. not crossing the fault) are not changing.”

4) For all baseline changes, slip rate (e.g., linear regression) and their error should be specified in the analysis part. It would prefer to show the table in supplementary material.

We inserted a Tab. S1 to the supplementary material. The table lists for each baseline the baseline length, inclination, azimuth, baseline change, strain and drift with drift uncertainty.

5) The method of integration for all baseline changes is unclear. Please clarify the authors used only baselines across-fault, or also used baselines along-fault. Do they calculate a simple average when integrating all baselines?

See point 3 above.

6) Comments of the Istanbul earthquake hazard should be included in the discussion section. As ‘Istanbul’ is written in the title, readers would want to know the relationship between this study and it.

We added a paragraph to the Introduction, see comment to the editor on page 1.

There are some minor revisions.

7) While the authors present how to analyze (Line 110–141), it should not be ‘Discussion’ section. Analysis and discussion should be separated into two sections.

We changed the section title to “*Results and Discussion*” and included subheadings.

8) Fig.2: It would prefer to use the unit ‘mm/a’ instead of ‘mm’.

Done.

9) More explanation on pressure sensor’s drift should be given in Line 139–141. The authors should provide a reference why “uplift is unlikely”.

We rephrased the sentence about the likelihood of uplift and included a reference to Armijo et al. (2002) and Bécél, et al. (2010):

“Because the Sea of Marmara is an extensional step-over or pull-apart structure system including substantial subsidence³⁵ with the basement imaged at a depth of ~4.5 km below the Central High and therefore close to the geodetic network, uplift is unlikely³⁶”

We added to the caption of Supplementary Figure 2:

“In general, the pressure sensors used are known to have a long term mean drift of 0.88+/- 0.73 kPa/a¹. As a result, drift and the differences between the pressure measurements might all be explained with sensor drift. Resolution of pressure is around 20 Pa corresponding to an effective depth resolution of 20 mm.”

10) ‘Marmara Sea’ should be replaced by ‘Sea of Marmara’ throughout the manuscript (e.g., Line 28).

We change “*Marmara Sea*” to “*Sea of Marmara*” throughout the manuscript, but not in the list of references.

I hope these comments will be helpful.

We highly appreciate the helpful, constructive and positive comments.

Reviewers' comments:

Reviewer #1 (Remarks to the Author):

Reviewer #1

Round 2

Interseismic Strain Build-up on the Submarine North Anatolian Fault Offshore Istanbul
by D. Lange et al.

I find my points thoroughly addressed by the authors in the revised version of the manuscript and appreciate the additions and modifications made. Particularly the quantitative analysis on expectable strain rates across the fault with consideration of variable locking depth, rigidity and sediment thickness greatly facilitates the assessment of the significance of the data.

I have one question on the analysis. Is the assumption of a steady fault slip rate of 20 mm/a below the locking depth critical in the sense that lower slip rates, as have been also proposed for this fault segment, would result in strain (rates) close to or even below the sensitivity of the network?

If so, I would appreciate a note on that to make it clear what the data really can show and where are the limits when it comes to conclusions. That being said I agree with the view that a locked state is very likely on this segment of the NAF.

Small things

Line 173 Should be Fig. 3e instead of 4d?

Line 201 processes

Lines 219, 222, 231, 251, Figure 5: Is it strain or strain rate? If it is the former: is it the strain accumulated during the period of the measurements? If it is the latter: strain rate is not dimensionless as stated in Figure 5. Note also, that in Figure 5 the strain sensitivity of the network is directly compared to expectable strain rates.

Line 276 "in the vicinity of the locked fault"

Line 286 phases instead of phased?

Lines 304/315 centre/center

Line 534 the Sea of Marmara

Line 536 "repeated influx of cold water"

Supplement Line 173 GPa

Reviewer #3 (Remarks to the Author):

Interseismic Strain Build-up on the Submarine North Anatolian Fault Offshore Istanbul
By D. Lange et al.

I have read the authors' comment. I appreciate the authors to correct reviewer's demands and suggestions. The second version of the manuscript has been much improved. I think that construction of fault model, mention of earthquake hazard, and clarification of novelty are well made. I think this manuscript will be acceptable after some minor corrections have been done.

1) Please clarify the reason for setting the plate motion rate to 20 mm/a.

2) When showing modelled results, I think that it is easy to understand if there is a cartoon that shows the layer structure.

In this document, we address the reviewers' comments

Please find our answers below marked in **blue** (original comments marked in **black**).

We thank you and the reviewers for the constructive comments and suggestions.

Reviewers' comments:

Reviewer #1 (Remarks to the Author):

Round 2

Interseismic Strain Build-up on the Submarine North Anatolian Fault Offshore Istanbul
by D. Lange et al.

Dear Editor and authors,

I find my points thoroughly addressed by the authors in the revised version of the manuscript and appreciate the additions and modifications made. Particularly the quantitative analysis on expectable strain rates across the fault with consideration of variable locking depth, rigidity and sediment thickness greatly facilitates the assessment of the significance of the data.

I have one question on the analysis. Is the assumption of a steady fault slip rate of 20 mm/a below the locking depth critical in the sense that lower slip rates, as have been also proposed for this fault segment, would result in strain (rates) close to or even below the sensitivity of the network?

If so, I would appreciate a note on that to make it clear what the data really can show and where are the limits when it comes to conclusions. That being said I agree with the view that a locked state is very likely on this segment of the NAF.

We added this statement about our most conservative estimate for the locking depth:

“Modeling the minimal fault slip inferred from onshore geodetic observations (16 mm/a) results in 20% less strain and would still have been above to the sensitivity of the offshore geodetic network, in particular, due to the existence of weak shallow layers (Fig. 5a). From the modelling we find the locking depth of 3 km as the most conservative estimate.”

and

“The modelled strain is linearly dependent on the inferred slip rate of 20 mm/a³⁷.”

and in the Conclusions we modified the sentence about the resolution of the geodetic network:

“.....we show that the fault is locked down to at least 3 km and likely down to 5.5 km depth, into the crystalline basement. “

Small things

Line 173 Should be Fig. 3e instead of 4d?

We changed Fig. 4d to Fig. 3e.

Line 201 processes

Corrected.

Lines 219, 222, 231, 251, Figure 5: Is it strain or strain rate? If it is the former: is it the strain accumulated during the period of the measurements? If it is the latter: strain rate is not dimensionless as stated in Figure 5. Note also, that in Figure 5 the strain sensitivity of the network is directly compared to expectable strain rates.

We use strain rate and added “rate” to modified strain to “*strain rate*” and “*strain rate sensitivity*” to make this more clear. We corrected the dimension of the strain rate sensitivity (a^{-1}). In Figure 5 we use now the correct label “*strain rate sensitivity*”, so all the dimensions are comparable. Since we corrected this semantic wording issue the resolution of the locking depth remains unchanged.

Line 276 “in the vicinity of the locked fault”

Corrected, we inserted “*the*”.

Line 286 phases instead of phased?

Done.

Lines 304/315 centre/center

We use American English and changed “hypocentre” to “hypocenter”.

Line 534 the Sea of Marmara

Corrected.

Line 536 “repeated influx of cold water”

Done.

Supplement Line 173 Gpa

Done.

Reviewer #3 (Remarks to the Author):

Interseismic Strain Build-up on the Submarine North Anatolian Fault Offshore Istanbul

I have read the authors' comment. I appreciate the authors to correct reviewer's demands and suggestions. The second version of the manuscript has been much improved. I think that construction of fault model, mention of earthquake hazard, and clarification of novelty are well made. I think this manuscript will be acceptable after some minor corrections have been done.

1) Please clarify the reason for setting the plate motion rate to 20 mm/a.

We added this sentence about the range of slip rates from GPS and geological observations and dependency on strain on the slip rate:

"The slip rates estimated for the NAF from onshore observations range between 15 and 27 mm/a^{10,38,39} from GPS observations and between 15 and 19.7 mm/a from mass deposit considerations^{40,41}. We model fault creep with a rate of 20 mm/a (in-between the GPS and geological estimates) for different faulting depths, above which the fault is locked and below which it creeps. The modelled strain is linearly dependent on the inferred slip rate of 20 mm/a³⁷."

We added references documenting the choice of slip rate (GPS: Meade et al., 2002, mass deposit data/geological slip rate (Kurt et al., 2013, Grall et al., 2013).

As a result, the readers can now feasible assess the dependency of the modeled strain rate on the inferred slip rate shown in Figure 5.

2) When showing modelled results, I think that it is easy to understand if there is a cartoon that shows the layer structure.

We agree, and added a panel to Figure 5 showing the modelling concept with the modeled low rigidity layers and the vertical fault.